# Biological Activities and Secondary Metabolites from *Sophora tonkinensis* and Its Endophytic Fungi

**DOI:** 10.3390/molecules27175562

**Published:** 2022-08-29

**Authors:** Jia-Jun Liang, Pan-Pan Zhang, Wei Zhang, Da Song, Xin Wei, Xin Yin, Yong-Qiang Zhou, Xiang Pu, Ying Zhou

**Affiliations:** 1School of Basic Medicine, Guizhou University of Traditional Chinese Medicine, Guiyang 550025, China; 2School of Pharmacy, Guizhou University of Traditional Chinese Medicine, Guiyang 550025, China; 3School of Humanities and Management, Guizhou University of Traditional Chinese Medicine, Guiyang 550025, China

**Keywords:** *S. tonkinensis*, phytochemistry, pharmacology, review

## Abstract

The roots of *Sophora tonkinensis* Gagnep., a traditional Chinese medicine, is known as Shan Dou Gen in the Miao ethnopharmacy. A large number of previous studies have suggested the usage of *S. tonkinensis* in the folk treatment of lung, stomach, and throat diseases, and the roots of *S. tonkinensis* have been produced as Chinese patent medicines to treat related diseases. Existing phytochemical works reported more than 300 compounds from different parts and the endophytic fungi of *S. tonkinensis*. Some of the isolated extracts and monomer compounds from *S. tonkinensis* have been proved to exhibit diverse biological activities, including anti-tumor, anti-inflammatory, antibacterial, antiviral, and so on. The research progress on the phytochemistry and pharmacological activities of *S. tonkinensis* have been systematically summarized, which may be useful for its further research.

## 1. Introduction

*Sophora tonkinensis* Gagnep. belongs to the *Sophora* genus of the Leguminosae family, which is widely distributed in the southwest provinces of China [1,2]. As a famous folk medicine of the Miao people, the roots of *S. tonkinensis* were known as Shan Dou Gen or Guang Dou Gen in the Miao ethnopharmacy [3,4]. The early medicinal records of Shan Dou Gen were contained in the classics *“Kai Bao Ben Cao”*, in which *S. tonkinensis* showed the effect of anti-sore throat diseases [5,6]. A large number of previous studies have suggested the usage of *S. tonkinensis* in the folk treatment of upper respiratory tract infection, including lung and throat diseases. Meanwhile, *S. tonkinensis* is also highly effective in the treatment of liver and skin diseases [7,8]. Moreover, the roots of *S. tonkinensis* can also be combined with other medicines to form dozens of clinical and marketing Chinese patent medicines, such as *Kai Hou Jian throat spray, Shuyanqing Spray,* and *Watermelon Frost Spray*, which is usually used for treatment of pharyngitis, tonsillitis, and aphthous ulcers [9,10,11]. Existing phytochemical works reported more than 300 compounds with various structural skeleton types from different parts and endophytic fungi of *S. tonkinensis*. Some of the isolated monomer compounds from *S. tonkinensis* have been proved to exhibit diverse biological activities, including anti-tumor, anti-inflammatory, antibacterial, antiviral, and so on [12,13,14,15,16,17]. Herein, the research progress on the phytochemistry and pharmacological activities of *S. tonkinensis* have been systematically summarized, which may be useful for its further research.

## 2. Phytochemistry

Previous studies have shown that alkaloids, flavonoids, triterpenoids, and triterpenoid saponins were the main chemical components isolated from *S. tonkinensis*. To date, 78 (**1**–**78**) alkaloids, 115 (**79**–**193**) flavonoids, 46 (**194**–**239**) triterpenes and triterpenoid saponins, and 37 (**240**–**276**) other compounds have been isolated from *S. tonkinensis*, and it is worth mentioning that 40 (**277**–**316**) compounds were also isolated from the endophytic fungi produced by *S. Tonkinensis* (Table 1, Figure 1).

### 2.1. Alkaloids

The alkaloids isolated in *S. tonkinensis* were mainly quinolizidine-type alkaloids [73]. To date, 78 alkaloids have been identified and isolated, of which 49 (**1**–**49**) are matrine type alkaloids. Sophtonseedline A (**46**) was isolated from the seeds of *S. tonkinensis*, which featured an unprecedented 5/6/6/6 tetracyclic skeleton [19]. Meanwhile, tonkinensines A (**58**) and B (**59**) with the rare multi group bridging structures were isolated from *S. tonkinensis* also [25].

### 2.2. Flavonoids

Flavonoids generically referred to the compounds with C6-C3-C6 structure skeleton. The flavonoids were rich in *S. tonkinensis*, and more than 115 flavonoids have been reported as far as we know. Their structural types can be classified as flavonoids (**79****-87**), flavonols (**88**–**97**), isoflavones and dihydroisoflavones (**98**–**118**), dihydroflavones (**119**–**158**), chalcones and dihydrochalcones (**159**–**167**), pterostanes (**168**–**191**), and flavanols (**192**–**193**)**.** Interestingly, tonkinochromanes A (**143**) and B (**156**) may ring-fused in the isoprenyl substituents [53]. Meanwhile, sophoraflavones A (**87**) and B (**86**) were the rare 5-deoxyflavonoids from the roots of *S. tonkinensis* [32]. Among the eighteen flavonoids identified using UPLC-ESI-LTQ/MS methods, formononetin (**107**), quercetin (**88**), rutin (**96**), isoquercitrin (**94**), and quercitrin (**95**) were suggested as the major quality markers of *S. tonkinensis* roots [37].

### 2.3. Triterpenoids and Triterpenoid Saponins

As far as we know, more than 46 (**194**–**239**) triterpenoids and triterpenoid saponins have been isolated from *S. tonkinensis*. Isolated triterpenoids are mainly of the oleanane type with carbonyl substitution at position C-22 [30,74]. Compared with flavonoids and alkaloids, the triterpenoids and triterpenoid saponins of *S. tonkinensi*s were rarely reported [59,61,62].

### 2.4. Other Compounds

In addition to alkaloids, flavonoids, and triterpenoids, a total of 37 (**240**–**276**) phenolic acids, sterols, and other compounds were reported from *S.*
*tonkinensis*. Two new 2-arylbenzofuran dimers, shandougenines A (**263**) and B (**264**), were isolated from the roots of *S.*
*tonkinensis*. It is noteworthy that shandougenine A (**263**) has the unique dimeric 2-Arylbenzofuran with a C-3\C-5 bond, and shandougenine B (**264**) was the natural dimeric 2-arylbenzofuran with a novel C-3/C-3 bond [40]. Meanwhile, a new propenyl phenylacetone was also isolated from *S. tonkinensis* and named sophoratonin H (**257**) [42].

### 2.5. Compounds Produced by Endophytic Fungi

The endophytic fungus *Xylaria* sp.GDG-102, *Penicillium macrosclerotiorum*, *Penicillium vulpinum, Diaporthe* sp.GDG-118, and *Xylaria* sp. GDGJ-368 [65,66,69,71] were isolated from *S. tonkinensis*, and some compounds produced by these endophytic fungi were interesting. More than 40 (**277**–**316**) compounds have been isolated from its endophytic fungi. Xylapeptide A (**301**) identified from the associated fungus *Xylaria* sp. GDG-102 was the first example of cyclopentapeptide with an L-Pip of terrestrial origin [70].

## 3. Pharmacological Activities

### 3.1. Anti-Inflammatory Effect

Reported studies have shown the anti-inflammatory activities of *S. tonkinensis* (Table 2) **[45,75]**. Some novel compounds, including 12,13-dehydrosophoridine (**16**) from *S.*
*tonkinensis*, showed significant activity against inflammatory cytokines TNF-α and IL-6 on LPS-induced RAW264.7 macrophages [23]. Moreover, 6,8-diprenyl-7,4’-dihydroxyflavanone (DDF) (**1****19**) inhibited the production of NO and the expression of TNF-α, IL-1β, and IL-6 [45]. Meanwhile, the compounds 2′-hydroxyglabrol (**131**), glabrol (**121**), maackiain (**168**), and bolusanthin IV (**261**) showed strong inhibitory effects on IL-6 [47]. Sophotokin (**174**) dose-dependently inhibited the lipopolysaccharide (LPS)-stimulated production of NO, TNF-α, PGE_2_, and IL-1β in microglial cells [34]. Moreover, the orally administered roots extract of *S. tonkinensis* attenuated the total leukocytes, eosinophil infiltration, and IL-5 level in BAL fluids [76]. Another study also showed *S. tonkinensis* were able to reduce TNF-α, NO, and IL-6 contents in rat paw edema induced by carrageenan [77].

### 3.2. Anti-Tumor Effect

The anti-tumor effect was one of the most reported activities of *S.*
*tonkinensis* (Table 2). The chloroform extracts of *S.*
*tonkinensis* have been discovered its inhibitory effect on cell viability and clonal growth in a dose-dependent manner [87]. Meanwhile, the extracts of *S. tonkinensis* also have been reported the inhibit ability target the proliferation, adhesion, invasion, and metastasis of mouse melanoma cells [86]. The anticancer activities of compounds have also been reported [38]. The natural compounds from *S.*
*tonkinensis* exhibited inhibitory effects against different tumor cells. The growth-inhibitory and apoptosis-inducing activities of sophoranone (**120**) for leukemia U937 cells were investigated [88].

### 3.3. Hepatoprotective

The components of *S. tonkinensis* were reported significant protective effects against immune induced liver injury (Table 2). Previous works suggested that the nonalkaloid constituents of *S. tonkinensis* obviously reduced the alanine aminotransferase (ALT), aspartate aminotransferase (AST) serum, malondialdehyde (MDA), and nitric oxide (NO), as well as increased the superoxide dismutase (SOD) and glutathione (GSH) in mice with immune-induced liver injury [13]. The water extract of *S. tonkinensis* alleviated hepatic inflammation, liver fibrosis, and hepatic lipids accumulation [91]. Compounds matrine (**1**) and oxymatrine (**4**) may be the main components contributing to the lipid-lowering activity of the water extract of *S. tonkinensis* [91]. Meanwhile, two purified polysaccharide fractions (STRP1 and STRP2) from the roots of *S. tonkinensis* have been reported to attenuate hepatic oxidative damage *in vivo* [95]. In addition, some compounds, including sophocarpine (**34**) from *S. tonkinensis* have been reported to significantly improve liver injury in mice [93]. 

### 3.4. Anti-Viral Activity

The compounds isolated from *S. tonkinensis* (Table 2), such as 3-(4-Hydroxyphenyl)-4-(3-methoxy-4- hydroxyphenyl)-3,4-dehydroquinolizidine (**75**), cermizine C (**70**), jussiaeiine A (**68**), jussiaeiine B (**67**), (+)-5α-hydroxyoxysophocarpine (**17**), (−)-12*β*- hydroxyoxysophocarpine (**18**), and (−)-clathrotropine (**64**), have reported the anti-coxsackie virus B_3_ (CVB_3_) activities with IC_50_ values rang of 0.12~6.40 µmol/L [26]. The compounds sophtonseedline B (**188**) and (−)-trifolirhizin (**190**) from *S. tonkinensis* exhibited anti-tobacco mosaic virus (TMV) activities with the inhibition rates of 69.62% and 68.72%, respectively, at a concentration of 100 µg/mL [56]. The other compounds, including sophtonseedline D (**23**), sophtonseedline F (**8**)**,** and (−)-*N*-formylcytisine (**52**)**,** have been reported to have anti-TMV activities as well [19]. In addition to TMV, compounds (+)-oxysophocarpine (**20**), (−)-sophocarpine (**34**), and (−)-13,14-Dehydrosophoridine (**16**) have showed anti-HBV activities [20].

### 3.5. Anti-Antioxidant Activities

The antioxidant activities of chloroform, ethyl acetate, *N-*butanol, and ethanol extracts of *S. tonkinensis* have been tested (Table 2). The results of DPPH, ABTS, and OH radical scavenging assay showed that all extracts exhibited antioxidant activities [98]. Some compounds from *S. tonkinensis* exhibited antioxidant activities. It is noteworthy that shandougenine A (**263**), shandougenine C (**127**), shandougenine D (**128**)**,** and 7,4’-Dihydroxyisoflavone (**103**) showed stronger superoxide anion radical scavenging capacity than the known flavanone luteolin. Shandougenines B (**264**) showed DPPH free radical and ABTS cation radical scavenging capacity. Shandougenine A (**263**), shandougenine C (**127**), shandougenine D (**128**), bolusanthin IV (**261**), 2-(2’,4’-Dihydroxyphenyl)-5,6-methylenedioxybenzofuran (**260**), and demethylmedicarpin (**179**) were reported parallel ABTS cation radical scavenging capacity to the positive control [40].

### 3.6. Toxicity

The roots of *S. tonkinensis* were the famous toxic Miao drug (Table 2) and were named Shan Dou Gen or Guang Dou Gen [4,110]. The aqueous and alcoholic parts of *S. tonkinensis* caused obvious liver damage in mice, which could result in both the alteration of liver function and the organelle damage of hepatocytes [111,112]. Meanwhile, the extracts of *S. tonkinensis* exhibited pulmonary toxicity, which may trigger pulmonary cancer, dyspnea, and oxidative stress [113]. The obvious toxicity of sophoranone (**120**) to *zebrafish* was mainly characterized as hepatotoxicity, neurotoxicity, cardiovascular toxicity, and nephrotoxicity in the acute toxicity model [104]. Besides, the alkaloids matrine (**1**), oxymatrine (**4**), cytisine (**50**), and sophocarpine (**34**) of *S. tonkinensis* showed significant cardiotoxicity [114].

### 3.7. Other Pharmacological Activities

The extracts of *S. tonkinensis* have the ability to reduce blood glucose and resist microbial activities (Table 2, Figure 2). Cytochalasin E (**310**) and H (**306**) inhibit a variety of plant pathogens [71]. The flavonoid-rich extracts of *S. tonkinensis* administrated orally to mice significantly increased sensibility to insulin, as well as reduced fasting blood-glucose levels [33]. Moreover, matrine (**1**) from *S. tonkinensis* could improve glucose metabolism and increased insulin secretion in diabetic mice, which may be used as a potential drug for diabetes treatment [108]. Methanol extracts of *S. tonkinensis* exhibited antidiarrheal activities [115]. Moreover, diverse anti-microbial activities of compounds from *S. tonkinensis* and its endophytic fungi have been reported [26,67].

## 4. Conclusion and Future Prospective

In this review, we provide a detailed summary of the medicinal chemistry, pharmacological activities, and related toxicity research of *S. tonkinensis.* Structurally, more than 300 compounds have been isolated from *S. tonkinensis* and its endophytic fungi, including alkaloids, triterpenes and triterpenoid saponins, flavonoids, and so on. Some of the star molecules, including matrine (**1**) and oxymatrine (**4**), were documented to exhibit well biological activities [110]. For its pharmacological research, previous studies suggested the usage of *S. tonkinensis* in the folk treatment of upper respiratory tract infection diseases. It is generally believed that the alkaloid components of *S. tonkinensis* were the main active substances in the roots of *S. tonkinensis* [116]. Interestingly, the extracts of *S. tonkinensis* have been reported for hepatotoxicity, while the other related studies showed the opposite hepatoprotective effects. The in-depth toxicological or structure-activity relationship study may be worth for further research. Moreover, the roots of *S. tonkinensis* combined with other medicines form dozens of marketing Chinese patent medicine for the treatments of pharyngitis, tonsillitis, and aphthous ulcers [9,10,11]. However, it is rare for its prescription pharmacological research in the treatment of upper respiratory tract diseases, especially works on the drug combination mechanism, which may need to be further developed.

## Figures and Tables

**Figure 1 molecules-27-05562-f001:**
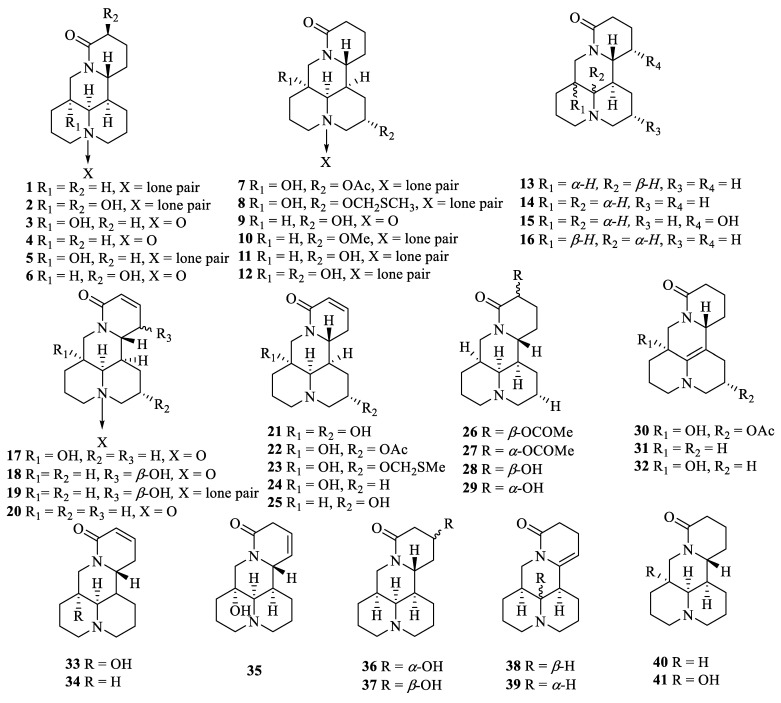
Structures of compounds **1****–316** from *S. tonkinensis*.

**Figure 2 molecules-27-05562-f002:**
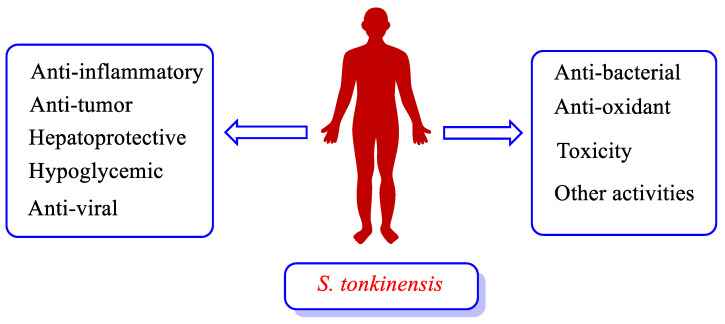
The biological activities of *S. tonkinensis*.

**Table 1 molecules-27-05562-t001:** The comprehensive list of the compounds from *S**. tonkinensis* and its Endophytic fungus.

NO	Compounds	Molecular Formula	Parts of Plant	References
**Matrine-Type alkaloids**				
**1**	Matrine	C_15_H_24_N_2_O	Roots	[12]
**2**	5*α*,14*β*-Dihydroxymatrine	C_15_H_24_N_2_O_3_	Roots	[12]
**3**	(+)-5*α*-Hydroxyoxymatrine	C_15_H_24_N_2_O_3_	Roots	[12]
**4**	(+)-Oxymatrine	C_15_H_24_N_2_O_2_	Roots	[18]
**5**	(+)-5*α*-Hydroxymatrine ((+)-Sophoranol)	C_15_H_24_N_2_O_2_	Roots	[12]
**6**	(−)- 14*β*-Hydroxyoxymatrine	C_15_H_24_N_2_O_3_	Roots	[18]
**7**	Sophtonseedline E	C_17_H_26_N_2_O_4_	Seeds	[19]
**8**	Sophtonseedline F	C_17_H_28_N_2_O_3_S	Seeds	[19]
**9**	Sophtonseedline G	C_15_H_24_N_2_O_3_	Seeds	[19]
**10**	Sophtonseedline H	C_16_H_26_N_2_O_2_	Seeds	[19]
**11**	(+)-9*α*-Hydroxymatrine	C_15_H_24_N_2_O_2_	Seeds	[19]
**12**	(+)-5*α*-9*α*-Dihydroxymatrine	C_15_H_24_N_2_O_3_	Seeds	[19]
**13**	(+)-Allomatrine (Sophoridine)	C_15_H_24_N_2_O	Roots	[20]
**14**	(+)-Lehmannine	C_15_H_24_N_2_O	Roots	[20]
**15**	(+)-12*α-*Hydroxysophocarpine	C_15_H_24_N_2_O_2_	Roots	[20]
**16**	(−)-13,14-Dehydrosophoridine (12,13-Dehydrosophoridine)	C_15_H_24_N_2_O	Roots	[20]
**17**	(+)-5*α-*Hydroxyoxysophocarpine	C_15_H_22_N_2_O_3_	Roots	[14]
**18**	(−)-12*β-*Hydroxyoxysophocarpine	C_15_H_22_N_2_O_3_	Roots	[14]
**19**	(−)-12*β-*Hydroxysophocarpine	C_15_H_22_N_2_O_2_	Roots	[14]
**20**	(+)-Oxysophocarpine	C_15_H_22_N_2_O_2_	Roots	[14]
**21**	Sophtonseedline B	C_15_H_22_N_2_O_3_	Seeds	[19]
**22**	Sophtonseedline C	C_17_H_24_N_2_O_4_	Seeds	[19]
**23**	Sophtonseedline D	C_17_H_26_N_2_O_3_S	Seeds	[19]
**24**	(−)-5*α*-Hydroxysophocarpine (13,14-Dehydrosophoranol)	C_15_H_22_N_2_O_2_	Seeds	[19]
**25**	(−)-9*α*-Hydroxysophocarpine	C_15_H_22_N_2_O_2_	Seeds	[19]
**26**	(−)-14*β-*Acetoxymatrine	C_17_H_26_N_2_O_3_	Leaves	[21]
**27**	(+)-14*α*-Acetoxymatrine	C_17_H_26_N_2_O_3_	Leaves	[21]
**28**	(−)-14*β*-Hydroxymatrine	C_15_H_24_N_2_O_2_	Leaves	[21]
**29**	(+)-14*α*-Hydroxymatrine	C_15_H_24_N_2_O_2_	Leaves	[21]
**30**	Sophtonseedline I	C_17_H_24_N_2_O_4_	Seeds	[19]
**31**	6,7-Dehydro-matrine	C_15_H_22_N_2_O	Seeds	[19]
**32**	5-Hydroxy-6,7-dehydro-matrine	C_15_H_22_N_2_O_2_	Seeds	[19]
**33**	(+)-13,14-Dehydrosophoranol	C_15_H_22_N_2_O_2_	Roots	[22]
**34**	(−)-Sophocarpine	C_15_H_22_N_2_O	Roots	[12]
**35**	(+)-5*α-*Hydroxylemannine	C_15_H_22_N_2_O_2_	Roots	[14]
**36**	13*α-*Hydroxymatrine	C_15_H_24_N_2_O_2_	Roots	[23]
**37**	13*β*-Hydroxymatrine	C_15_H_24_N_2_O_2_	Roots	[23]
**38**	11,12-Dehydroallmatrine	C_15_H_22_N_2_O	Roots	[1]
**39**	11,12-Dehydromatrine	C_15_H_22_N_2_O	Roots	[1]
**40**	(+)-Matrine *N*-oxide	C_15_H_24_N_2_O	Leaves	[21]
**41**	(+)-Sophoranol *N*-oxide	C_15_H_24_N_2_O_2_	Leaves	[21]
**42**	(+)-7,11-Dehydromatrine	C_15_H_22_N_2_O	Roots	[22]
**43**	Alopecurin A	C_15_H_22_N_2_O_4_	Seeds	[19]
**44**	Sophtonseedline J	C_15_H_20_N_2_O_3_	Seeds	[19]
**45**	Sophtonseedline K	C_15_H_20_N_2_O_3_	Seeds	[19]
**46**	Sophtonseedline A	C_15_H_22_N_2_O_2_	Seeds	[19]
**47**	5,6-Dehydro-matrine	C_15_H_22_N_2_O	Seeds	[19]
**48**	Isosophocarpine	C_15_H_22_N_2_O	Roots	[23]
**49**	(+)-Sophoramine (7*β*-Sophoramine)	C_15_H_20_N_2_O	Roots	[14]
**Cytisine-type alkaloids**				
**50**	(−)-Cytisine	C_11_H_14_N_2_O	Seeds	[19]
**51**	*N*-Methylcytisine	C_12_H_16_N_2_O	Seeds	[19]
**52**	(−)-*N*-Formylcytisine	C_12_H_14_N_2_O_2_	Seeds	[19]
**53**	*N*-Acylcytisine	C_13_H_16_N_2_O_2_	Seeds	[19]
**54**	(−)-*N*-Methylcytisine	C_12_H_16_N_2_O	Roots	[18]
**55**	(−)-*N*-Hexanoylcytisine	C_17_H_24_N_2_O_2_	Roots	[24]
**56**	(−)-*N*-Ethylcytisine	C_13_H_18_N_2_O	Roots	[24]
**57**	(−)-*N*-Propionylcytisine	C_14_H_18_N_2_O_2_	Roots	[24]
**58**	Tonkinensine A	C_28_H_26_N_2_O_6_	Roots	[25]
**59**	Tonkinensine B	C_28_H_26_N_2_O_6_	Roots	[25]
**Anagyrine-type alkaloids**				
**60**	17-Oxo-*α*-isosparteine	C_15_H_24_N_2_O	Leaves	[21]
**61**	(−)-Anagyrine	C_15_H_20_N_2_O	Roots	[12]
**62**	(−)-Thermopsine	C_15_H_20_N_2_O	Roots	[12]
**63**	(−)-Baptifoline	C_15_H_20_N_2_O_2_	Leaves	[21]
**64**	(−)-Clathrotropine	C_17_H_22_N_2_O_4_	Roots	[26]
**65**	Lanatine A	C_22_H_29_N_3_O_3_	Roots	[26]
**Lupine-types and other alkaloids**				
**66**	Lamprolobine	C_15_H_24_N_2_O_2_	Leaves	[21]
**67**	Jussiaeiine B	C_16_H_24_N_2_O_2_	Roots	[26]
**68**	Jussiaeiine A	C_13_H_20_N_2_O_2_	Roots	[26]
**69**	Senepodine H	C_14_H_26_NO^+^	Roots	[26]
**70**	Cermizine C	C_11_H_21_N	Roots	[26]
**71**	Senepodine G	C_11_H_20_N^+^	Roots	[26]
**72**	Harmine	C_13_H_12_N_2_O	Roots	[1]
**73**	Tonkinensine C	C_16_H_16_N_2_O_2_	Roots	[1]
**74**	Perlolyrine	C_16_H_12_N_2_O_2_	Roots	[1]
**75**	3-(4-Hydroxyphenyl)-4-(3-methoxy-4-hydroxyphenyl)-3,4-dehydroquinolizidine	C_22_H_25_NO_3_	Roots	[26]
**76**	1-(6,7-dihydro-5H-pyrrolo[1,2-*a*]imidazol-3-yl)ethanone	C_8_H_10_N_2_O	Roots	[27]
**77**	Cyclo (Pro-Pro)	C_10_H_14_N_2_O_2_	Roots	[27]
**78**	Nicotinic acid	C_6_H_5_NO_2_	Roots	[27]
**Flavonoids**				
**79**	4′,7-Dihydroxyflavone	C_15_H_10_O_4_	Roots	[28]
**80**	Wogonin	C_16_H_12_O_5_	Roots	[29]
**81**	Luteolin	C_15_H_10_O_4_	Roots	[29]
**82**	Luteolin-7-glucoside	C_21_H_20_O_11_	Roots	[30]
**83**	Baicalein 7-O-*β*-_Ⅾ_-glucuronide	C_21_H_18_O_11_	Roots	[31]
**84**	Bayin	C_21_H_20_O_9_	Roots	[15]
**85**	Swertisin	C_22_H_22_O_10_	Roots	[31]
**86**	Sophoraflavone B	C_21_H_20_O_9_	Roots	[32]
**87**	Sophoraflavone A	C_27_H_30_O_13_	Roots	[32]
**Flavonols**				
**88**	Quercetin	C_15_H_10_O_7_	Roots	[33]
**89**	Morin	C_15_H_10_O_7_	Roots	[31]
**90**	6,8-Diprenylkaempferol	C_25_H_26_O_6_	Roots	[34]
**91**	8-C-prenylkeamferol	C_20_H_18_O_6_	Roots	[35]
**92**	Dehydrolupinifolinol	C_25_H_24_O_6_	Roots	[33]
**93**	Tonkinensisol	C_25_H_24_O_6_	Roots	[15]
**94**	Isoquercitrin	C_21_H_20_O_12_	Roots	[36]
**95**	Quercitrin	C_21_H_20_O_11_	Roots	[37]
**96**	Rutin (Quercetin-3-O-*β-_D_*-rutinoside)	C_27_H_30_O_16_	Roots	[31]
**97**	Isorhamnetin-3-O-*β-_D_-*rutinoside	C_28_H_32_O_16_	Roots	[31]
**Isoflavones and Dihydroisoflavones**				
**98**	8,4′-Dihydroxy-7-methoxyisoflavone	C_16_H_12_O_5_	Roots	[38]
**99**	5,7,2′,4′-Tetrahydroxyisoflavone	C_15_H_10_O_6_	Roots	[38]
**100**	Calycosin	C_16_H_12_O_5_	Roots	[38]
**101**	7,3′-Dihydroxy-5’-methoxyisoflavone	C_16_H_12_O_5_	Roots	[38]
**102**	7,4′-Dihydroxy-3′-methoxyisoflavone	C_16_H_12_O_5_	Roots	[38]
**103**	Daidzein (7,4’-Dihydroxyisoflavone)	C_15_H_10_O_4_	Roots	[38]
**104**	7,3′-Dihydroxy-8,4′-dimethoxyisoflavone	C_17_H_14_O_6_	Roots	[38]
**105**	7,8-Dihydroxy-4′-methoxyisoflavone	C_16_H_12_O_5_	Roots	[38]
**106**	7,3′,4′-Trihydroxyisoflavone	C_15_H_10_O_5_	Roots	[38]
**107**	Formononetin	C_16_H_12_O_4_	Roots	[39]
**108**	Genistein	C_15_H_10_O_5_	Roots	[39]
**109**	Wighteone	C_20_H_18_O_5_	Roots	[40]
**110**	8-Methylretusin	C_17_H_14_O_5_	Roots	[41]
**111**	7-Methoxyebenosin	C_22_H_22_O_4_	Roots	[42]
**112**	Tectorigenin	C_16_H_12_O_6_	Roots	[43]
**113**	Butesuperin A	C_26_H_22_O_8_	Roots	[44]
**114**	Butesuperin B -7′-O-*β*-glucopyranoside	C_33_H_34_O_14_	Roots	[44]
**115**	Genistin	C_21_H_20_O_10_	Roots	[33]
**116**	Ononin (Formononetin-7-O-*β*-_D_-glucoside)	C_22_H_22_O_9_	Roots	[33]
**117**	Daidzein-4′-glucoside-rhamnoside	C_27_H_30_O_13_	Roots	[37]
**118**	Sophorabioside	C_27_H_30_O_14_	Roots	[37]
**Dihydroflavones**				
**119**	6,8-Diprenyl-7,4′-Dihydroxyflavanone	C_25_H_28_O_4_	Roots	[45]
**120**	Sophoranone	C_30_H_36_O_4_	Roots	[45]
**121**	Glabrol	C_25_H_28_O_4_	Roots	[45]
**122**	6,8-Diprenyl-7,2′,4′-trihydroxyflavanone	C_25_H_28_O_5_	Roots	[45]
**123**	Lespeflorin B_4_	C_30_H_36_O_6_	Roots	[33]
**124**	(2*S*)-7,4′-Dihydroxy-5′-aldehyde-8,3′-(3′′-methylbut-2′′-enyl) flavanone	C_26_H_28_O_5_	Roots	[34]
**125**	(2*S*)-7,2′,4′-Trihydroxy-8,3′,5′-(3′′-methyl- but-2′′-enyl) flavanone	C_30_H_36_O_5_	Roots	[34]
**126**	Tonkinochromane J	C_25_H_28_O_5_	Roots	[46]
**127**	Shandougenine C	C_30_H_36_O_5_	Roots	[40]
**128**	Shandougenine D	C_25_H_28_O_5_	Roots	[40]
**129**	Sophoratonin F	C_35_H_44_O_4_	Roots	[42]
**130**	Lonchocarpol A	C_25_H_28_O_5_	Roots	[42]
**131**	2′-Hydroxyglabrol	C_25_H_28_O_5_	Roots	[47]
**132**	8,5′-Diprenyl-7,2′,4′-trihydroxyflavanone	C_25_H_28_O_5_	Roots	[45]
**133**	Sophoratonin A	C_27_H_28_O_4_	Roots	[42]
**134**	Sophoratonin B	C_30_H_32_O_4_	Roots	[42]
**135**	Tonkinochromane I	C_30_H_36_O_5_	Roots	[35]
**136**	Tonkinochromane G	C_30_H_36_O_5_	Roots	[34]
**137**	Sophoratonin C	C_30_H_30_O_4_	Roots	[42]
**138**	Sophoratonin D	C_30_H_36_O_4_	Roots	[42]
**139**	Flemichin D	C_25_H_26_O_5_	Roots	[45]
**140**	5-Dehydroxylupinifolin	C_25_H_26_O_4_	Roots	[34]
**141**	Lupinifolin	C_25_H_26_O_5_	Roots	[40]
**142**	2-(2′,4′-Dihydroxyphenyl)-8,8-dimethyl-1′-(3-methyl-2-butenyl)-8H-pyrano[2,3-d] chroman-4-one	C_25_H_26_O_5_	Roots	[48]
**143**	Tonkinochromane A	C_30_H_36_O_4_	Roots	[45]
**144**	Sophoranochromene	C_30_H_34_O_4_	Roots	[33]
**145**	2-[{2-(1-Hydroxy-1-methylethyl)-7-(3-methyl-2-butenyl)-2′,3-dihydrobenzofuran}-5-yl]-7-hydroxy-8-(3-methyl-2-butenyl)-chroman-4-one	C_30_H_36_O_5_	Roots	[49]
**146**	Sophoratonin E	C_30_H_32_O_4_	Roots	[42]
**147**	Tonkinochromane D	C_30_H_38_O_5_	Roots	[50]
**148**	Tonkinochromane E	C_32_H_42_O_5_	Roots	[50]
**149**	2-[{2′-(1-Hydroxy-1-methylethyl)-7′-(3-methyl-2-butenyl)-2′,3′-dihydrobenzofuran}-5′-yl]-7-hy-droxy-8-(3-methyl-2-butenyl) chroman-4-one	C_30_H_36_O_5_	Whole	[51]
**150**	Euchrenone A_2_	C_25_H_26_O_5_	Roots	[33]
**151**	Sophoratonin G	C_27_H_28_O_4_	Roots	[42]
**152**	Tonkinochromane K	C_30_H_36_O_6_	Roots	[46]
**153**	2-[{3′-Hydroxy-2′,2′-dimethyl-8′-(3-methyl-2-butenyl)} chroman-6′-yl]-7-hydroxy-8-(3-methyl-2-butenyl)-chroman-4-one	C_30_H_36_O_5_	whole	[51]
**154**	2-[{3-Hydroxy-2′,2-dimethyl-8-(3-methyl-2-butenyl)} chroman-6-yl]-7-hydroxy-8-(3-methyl-2-butenyl)-chro-man-4-one	C_31_H_38_O_4_	Roots	[49]
**155**	Tonkinochromane H	C_30_H_34_O_5_	Roots	[52]
**156**	Tonkinochromane B	C_30_H_36_O_4_	Roots	[53]
**157**	Kushenol E	C_25_H_28_O_6_	Roots	[46]
**158**	Naringenin 7-O-neo-hesperidoside	C_27_H_32_O_14_	Roots	[31]
**Chalcones and Dihydrochalcones**				
**159**	Isoliquiritigenin	C_15_H_12_O_4_	Roots	[47]
**160**	Sophoradin	C_30_H_36_O_4_	Roots	[34]
**161**	Xanthohumol	C_21_H_22_O_5_	Roots	[54]
**162**	7,9,2,4-Tetrahydroxy-8-isopentenyl-5-methoxychalcone	C_21_H_22_O_6_	Roots	[54]
**163**	Tonkinochromane C	C_28_H_30_O_4_	Roots	[53]
**164**	Tonkinochromane F	C_32_H_42_O_5_	Roots	[50]
**165**	Kuraridine	C_26_H_30_O_6_	Roots	[54]
**166**	Sophoradochromene	C_30_H_34_O_4_	Roots	[42]
**167**	Tonkinochromane L	C_21_H_24_O_4_	Roots	[46]
**Pterostanes**				
**168**	(−)-Maackiain	C_16_H_12_O_5_	Roots	[33]
**169**	Pisatin	C_17_H_14_O_6_	Roots	[39]
**170**	Maackiain-3-O-glucoside 6′’-acetate	C_24_H_24_O_11_	Roots	[47]
**171**	(−)-Maackiain 3-sulfate	C_16_H_11_O_8_S	Roots	[55]
**172**	*6aR,11aR*-1-hydroxy-4-isoprenyl-maackiain	C_21_H_20_O_6_	Roots	[48]
**173**	(*6aR,11aR*) - 2-hydroxy-3-methoxy-1-isopentenyl- maackiain	C_22_H_22_O_6_	Roots	[47]
**174**	Sophotokin	C_21_H_20_O_6_	Roots	[34]
**175**	(−)-Pterocarpin	C_17_H_14_O_5_	Seeds	[56]
**176**	Medicarpin	C_16_H_14_O_4_	Roots	[39]
**177**	*(6aR, 11aR)-*3-O-*β*-_D_-Glucopyranosylmedicarpin	C_22_H_24_O_9_	Roots	[24]
**178**	Medicarpin-3-O-glucoside 6″-acetate	C_24_H_26_O_10_	Roots	[47]
**179**	Demethylmedicarpin	C_15_H_12_O_4_	Roots	[40]
**180**	Homopterocarpin	C_17_H_16_O_4_	Roots	[42]
**181**	Dehydromaackiain	C_16_H_10_O_5_	Roots	[42]
**182**	Flemichapparin B	C_17_H_12_O_5_	Roots	[42]
**183**	Maackiapterocarpan B	C_21_H_18_O_6_	Roots	[57]
**184**	3-Methylmaackiapterocarpan B	C_22_H_20_O_6_	Roots	[47]
**185**	Erybraedin D	C_25_H_26_O_4_	Roots	[42]
**186**	Maackiapterocarpan A	C_21_H_20_O_6_	Roots	[42]
**187**	Medicagol	C_16_H_8_O_6_	Seeds	[56]
**188**	Sophtonseedlin B	C_28_H_28_O_13_	Seeds	[56]
**189**	Sophoratonkin	C_26_H_26_O_11_	Roots	[28]
**190**	(−)-Trifolirhizin	C_22_H_22_O_10_	Seeds	[56]
**191**	(−)-Trifolirhizin-6′′-monoacetate	C_24_H_24_O_11_	Seeds	[56]
**Flavanols**				
**192**	7,2’-Dihydroxy-4’-methoxy-isofiavanol	C_16_H_16_O_5_	Roots	[58]
**193**	(*3S,4R*)-4-hydroxy-7,4′-dimethoxyisoflavan 3′-O-*β*-_D_-glucopyranoside	C_23_H_28_O_10_	Roots	[24]
**Triterpenoids and Triterpenoid saponins**				
**194**	Subprogenin A	C_30_H_48_O_4_	Roots	[59]
**195**	Subprogenin B	C_30_H_48_O_5_	Roots	[59]
**196**	Subprogenin C	C_30_H_46_O_4_	Roots	[59]
**197**	Subprogenin C methylester	C_31_H_48_O_4_	Roots	[59]
**198**	Subprogenin D	C_30_H_46_O_4_	Roots	[59]
**199**	Subprogenin D methylester	C_31_H_48_O_4_	Roots	[59]
**200**	Abrisapogenol H	C_30_H_48_O_3_	Roots	[59]
**201**	Wistariasapogenol A	C_30_H_48_O_4_	Roots	[59]
**202**	Melilotigenin	C_30_H_46_O_5_	Roots	[59]
**203**	Abrisapogenol I	C_30_H_46_O_5_	Roots	[59]
**204**	Sophoradiol	C_30_H_50_O_2_	Roots	[59]
**205**	Cantoniensistiol	C_30_H_50_O_3_	Roots	[59]
**206**	Soyasapogenol B	C_30_H_50_O_3_	Roots	[59]
**207**	Soyasapogenol A	C_30_H_50_O_4_	Roots	[59]
**208**	Abrisapogenol C	C_30_H_50_O_4_	Roots	[59]
**209**	Abrisapogenol D	C_30_H_50_O_3_	Roots	[59]
**210**	Abrisapogenol E	C_30_H_50_O_4_	Roots	[59]
**211**	Kudzusapogenol A	C_30_H_50_O_5_	Roots	[59]
**212**	Abrisapogenol A	C_30_H_50_O_3_	Roots	[59]
**213**	Lupeol	C_30_H_50_O	Roots	[60]
**214**	Stigmasterol	C_29_H_48_O	Roots	[60]
**215**	*β*-Sitosterol	C_29_H_50_O	Roots	[60]
**216**	Daucosterol	C_35_H_60_O_6_	Roots	[60]
**217**	Subproside Ⅰ	C_48_H_78_O_19_	Roots	[61]
**218**	Subproside Ⅰ methylester	C_49_H_80_O_19_	Roots	[61]
**219**	Subproside Ⅱ	C_47_H_76_O_19_	Roots	[61]
**220**	Subproside Ⅱ methylester	C_48_H_78_O_19_	Roots	[61]
**221**	Soyasaponin A_3_ methylester	C_49_H_80_O_19_	Roots	[62]
**222**	Kuzusapogenol A methylester	C_49_H_80_O_20_	Roots	[62]
**223**	Soyasaponin I methylester	C_49_H_80_O_18_	Roots	[62]
**224**	Kaikasaponin Ⅲ methylester	C_49_H_80_O_17_	Roots	[62]
**225**	Soyasaponin Ⅱ methylester	C_48_H_78_O_17_	Roots	[62]
**226**	Kaikasapomn I methylester	C_49_H_80_O_17_	Roots	[62]
**227**	Kudzusaponin A_3_	C_47_H_76_O_19_	Roots	[61]
**228**	Soyasaponin II	C_47_H_76_O_17_	Roots	[61]
**229**	Dehydrosoyasaponin I	C_48_H_76_O_18_	Roots	[61]
**230**	Subproside Ⅶ	C_59_H_96_O_27_	Roots	[63]
**231**	Subproside Ⅶ methylester	C_60_H_98_O_27_	Roots	[63]
**232**	Subproside Ⅳ	C_54_H_88_O_23_	Roots	[63]
**233**	Subproside Ⅳ methylester	C_55_H_90_O_23_	Roots	[63]
**234**	Subproside Ⅴ	C_54_H_88_O_24_	Roots	[63]
**235**	Subproside Ⅴ methylester	C_55_H_90_O_24_	Roots	[63]
**236**	Subproside Ⅲ	C_54_H_86_O_24_	Roots	[61]
**237**	Subproside Ⅲ methylester	C_55_H_88_O_24_	Roots	[61]
**238**	Subproside Ⅵ	C_54_H_88_O_24_	Roots	[63]
**239**	Subproside Ⅵ methylester	C_55_H_90_O_24_	Roots	[63]
**Other compounds**				
**240**	Tyrosol	C_8_H_10_O_2_	Roots	[64]
**241**	4-(3-Hydroxypropyl) phenol	C_9_H_12_O_2_	Roots	[64]
**242**	Vanillin alcohol	C_8_H_10_O_3_	Roots	[64]
**243**	(±)-4-(2-Hydroxypropyl) phenol	C_9_H_12_O_2_	Roots	[64]
**244**	3,4,5-Trihydroxybenzoic acid	C_7_H_6_O_5_	Roots	[31]
**245**	3,4-Dihydroxybenzoic acid	C_7_H_6_O_4_	Roots	[31]
**246**	4-Hydroxy-3-methoxybenzoic acid	C_8_H_8_O_4_	Roots	[31]
**247**	*p*-Hydroxybenzonic acid	C_7_H_6_O_3_	Roots	[31]
**248**	Venillic acid	C_8_H_8_O_4_	Roots	[41]
**249**	*p*-Methoxybenzonic acid	C_8_H_8_O_3_	Roots	[27]
**250**	Salicylic acid	C_7_H_6_O_3_	Roots	[43]
**251**	Benzamide	C_7_H_7_NO	Roots	[64]
**252**	4-Methoxybenzamide	C_8_H_9_NO_2_	Roots	[64]
**253**	Docosyl caffeate	C_31_H_52_O_4_	Roots	[4]
**254**	Maltol	C_6_H_6_O_3_	Roots	[41]
**255**	(±)-3-( *p*-Methoxyphenyl) -1,2-propanediol	C_9_H_12_O_4_	Roots	[64]
**256**	3,4-Dimethoxybenzeneacrylic acid methyl ester	C_12_H_14_O_4_	Roots	[39]
**257**	Sophoratonin H	C_22_H_26_O_5_	Roots	[42]
**258**	Piscidic acid monoethyl ester	C_13_H_16_O_7_	Roots	[41]
**259**	2′,4′, 7-trihydroxy-6,8-bis(3-methyl-2-butenyl) flavanone	C_25_H_28_O_5_	Roots	[40]
**260**	2-(2′, 4′-dihydroxylphenyl)-5,6-methylenedioxybenzoftiran	C_15_H_10_O_5_	Roots	[56]
**261**	bolusanthin IV	C_15_H_12_O_4_	Roots	[40]
**262**	7,2′-Dihydroxy-4′,5′-methylenedioxyisoflavan	C_16_H_14_O_5_	Roots	[40]
**263**	Shandougenine A	C_30_H_18_O_10_	Roots	[40]
**264**	Shandougenine B	C_30_H_18_O_10_	Roots	[40]
**265**	(−)-Syringaresinol-4,4’-di-O-*β-_D_*-glucopyranoside	C_34_H_46_O_18_	Roots	[27]
**266**	(−)-Syringaresinol-4-O-*β*-_D_-glucopyranoside	C_28_H_36_O_13_	Roots	[27]
**267**	(−)-Pinoresinol-4,4’-di-O-*β-_D_*-glucopyranoside	C_32_H_42_O_16_	Roots	[27]
**268**	Pinoresinol	C_20_H_22_O_6_	Roots	[28]
**269**	Syringaresinol	C_22_H_26_O_8_	Roots	[28]
**270**	Medioresinol	C_21_H_24_O_7_	Roots	[28]
**271**	Coniferin	C_16_H_22_O_8_	Roots	[27]
**272**	4-Hydroxymethyl-2,6-dimethoxyphenol-1-O-*β*-*_D_*-glucopyranoside	C_15_H_22_O_9_	Roots	[27]
**273**	Syringin	C_17_H_24_O_9_	Roots	[29]
**274**	Sophtonseedlin A	C_23_H_14_O_9_	Roots	[56]
**275**	(6*S*,9*R*) -Roseoside	C_19_H_30_O_8_	Roots	[27]
**276**	(−)-Secoisolariciresinol-4-O-*β*-*_D_*-glucopyranoside	C_25_H_33_NO_9_	Roots	[27]
**Compounds produced by endophytic fungi**				
**277**	2-Methoxy-6-methyl-1,4-benzoquinone	C_8_H_8_O_3_	Endophytic Fungus *Xylaria* sp. GDG-102	[65]
**278**	1-Methyl emodin	C_16_H_12_O_5_	Endophytic Fungus *Penicillium macrosclerotiorum*	[66]
**279**	Isorhodoptilometrin	C_17_H_14_O_6_	Endophytic Fungus *Penicillium macrosclerotiorum*	[66]
**280**	(−)-5-Carboxylmellein	C_11_H_10_O_5_	Endophytic Fungus *Xylaria* sp. GDG-102	[65]
**281**	(−)-5-Methylmellein	C_11_H_12_O_3_	Endophytic Fungus *Xylaria* sp. GDG-102	[67]
**282**	Xylariphilone	C_11_H_16_O_4_	Endophytic Fungus *Xylaria* sp. GDG-102	[65]
**283**	Xylarphthalide A	C_11_H_10_O_6_	Endophytic Fungus *Xylaria* sp. GDG-102	[65]
**284**	2-Anhydromevalonic acid	C_6_H_10_O_3_	Endophytic Fungus *Xylaria* sp. GDG-102	[65]
**285**	(2*S*,5*R*)-2-Ethyl-5-methylhexanedioic acid	C_9_H_16_O_4_	Endophytic Fungus *Xylaria* sp. GDG-102	[65]
**286**	6-Heptanoyl-4-methoxy-2H-pyran-2-one	C_13_H_18_O_4_	Endophytic Fungus *Xylaria* sp. GDG-102	[65]
**287**	Xylareremophil	C_15_H_18_O_3_	Endophytic Fungus *Xylaria* sp. GDG-102	[68]
**288**	1*α*,10*α*-Epoxy-13-hydroxyeremophil-7(11)-en-12,8-*β*-olide	C_15_H_20_O_4_	Endophytic Fungus *Xylaria* sp. GDG-102	[68]
**289**	1*α*,10*α*-Epoxy-3*α*-hydroxyeremophil-7(11)-en-12,8*-β*-olide	C_15_H_20_O_5_	Endophytic Fungus *Xylaria* sp. GDG-102	[68]
**290**	Mairetolide B	C_15_H_20_O_4_	Endophytic Fungus *Xylaria* sp. GDG-102	[68]
**291**	Mairetolide G	C_15_H_22_O_5_	Endophytic Fungus *Xylaria* sp. GDG-102	[68]
**292**	1*β*,10*α*,13-Trihydroxyeremophil-7(11)-en-12,8-olide	C_16_H_24_O_4_	Endophytic Fungus *Xylaria* sp. GDG-102	[65]
**293**	(−)-3-Carboxypropyl-7-hydroxyphthalide	C_12_H_12_O_5_	Endophytic fungus *Penicillium vulpinum*	[69]
**294**	(−)-3-Carboxypropyl-7-hydroxyphthalide methyl ester	C_13_H_14_O_5_	Endophytic fungus *Penicillium vulpinum*	[69]
**295**	Sulochrin	C_17_H_16_O_7_	Endophytic fungus *Penicillium macrosclerotiorum*	[66]
**296**	Monoacetylasterric acid	C_18_H_16_O_9_	Endophytic fungus *Penicillium macrosclerotiorum*	[66]
**297**	Methyl dichloroasterrate	C_18_H_16_Cl_2_O_8_	Endophytic Fungus *Penicillium macrosclerotiorum*	[66]
**298**	Penicillither	C_18_H_17_ ClO_8_	Endophytic fungus *Penicillium macrosclerotiorum*	[66]
**299**	Methyl asterrate	C_18_H_18_O_8_	Endophytic fungus *Penicillium macrosclerotiorum*	[66]
**300**	Asterric acid	C_17_H_16_O_8_	Endophytic fungus *Penicillium macrosclerotiorum*	[66]
**301**	Xylapeptide A	C_30_H_45_N_5_O_5_	Endophytic Fungus *Xylaria* sp. GDG-102	[70]
**302**	Xylapeptide B	C_29_H_43_N_5_O_5_	Endophytic Fungus *Xylaria* sp. GDG-102	[70]
**303**	21-Acetoxycytochalasin J_2_	C_30_H_37_NO_4_	Endophytic fungus *Diaporthe* sp.GDG-118	[71]
**304**	21-Acetoxycytochalasin J_3_	C_30_H_39_NO_3_	Endophytic fungus *Diaporthe* sp.GDG-118	[71]
**305**	Cytochalasin J_3_	C_32_H_41_NO_4_	Endophytic fungus *Diaporthe* sp.GDG-118	[71]
**306**	Cytochalasin H	C_30_H_39_NO_5_	Endophytic fungus *Diaporthe* sp.GDG-118	[71]
**307**	7-Acetoxycytochalasin H	C_32_H_41_NO_6_	Endophytic fungus *Diaporthe* sp.GDG-118	[71]
**308**	Cytochalasin J	C_28_H_37_NO_4_	Endophytic fungus *Diaporthe* sp.GDG-118	[71]
**309**	Geomycin A	C_35_H_32_O_15_	Endophytic fungus *Penicillium macrosclerotiorum*	[66]
**310**	Cytochalasin E	C_28_H_33_NO_7_	Endophytic fungus *Diaporthe* sp.GDG-118	[71]
**311**	Cytochalasin K	C_28_H_33_NO_7_	Endophytic fungus *Xylaria* sp. GDG-102	[65]
**312**	Diaporthein B	C_20_H_28_O_6_	Endophytic fungus *Xylaria* sp. *GDGJ*-368	[72]
**313**	Piliformic	C_11_H_18_O_4_	Endophytic fungus *Xylaria* sp. *GDGJ*-368	[72]
**314**	Cytochalasin C	C_30_H_37_NO_6_	Endophytic fungus *Xylaria* sp. *GDGJ*-368	[72]
**315**	Cytochalasin D	C_30_H_37_NO_6_	Endophytic fungus *Xylaria* sp. *GDGJ*-368	[72]
**316**	(22E)-ergosta-6,22-diene-3*β*,*5β*,8*α*-triol	C_28_H_46_O_3_	Endophytic fungus *Xylaria* sp. *GDGJ*-368	[72]

**Table 2 molecules-27-05562-t002:** The comprehensive list of the pharmacological activities from *S. tonkinensis*.

Detail	Extracts/Compounds	*In Vivo/In Vitro*	Active Concentration/Dose	References
**Anti-inflammatory activity**				
Reduce TNF-*α*	(−)-Anagyrine (**61**)	In vitro	50 µM	[12]
	Sophocarpine (**34**)	In vitro	50 µM	[12]
	14*β-*Hydroxymatrine (**28**)	In vitro	50 µM	[12]
	7*β*-Sophoramine (**49**)	In vitro	50 µM	[12]
	Matrine (**1**)	In vivo	50 µM	[12]
	(+)-5*α*-Hydroxymatrine (**5**)	In vivo	50 µM	[12]
	12,13-Dehydrosophoridine (**16**)	In vitro	50 µM	[23]
	13*α*-Hydroxymatrine (**36**)	In vitro	50 µM	[23]
	13*β*-Hydroxymatrine (**37**)	In vitro	50 µM	[23]
	Isosophocarpine (**48**)	In vitro	50 µM	[23]
	Sophoridine (**13**)	In vitro	50 µM	[23]
	Water extract of roots	In vivo	0.3 g/kg	[75]
Inhibit the production of NO	sophoratonkin (**189**)	In vitro	IC_50_ *=* 33.0 µM	[28]
	Maackiain (**168**)	In vitro	IC_50_ *=* 27.0 µM	[28]
	Sophoranone (**120**)	In vitro	IC_50_ *=* 28.1 µM	[28]
	Sophoranochromene (**144**)	In vitro	IC_50_ *=* 13.6 µM	[28]
	Tonkinochromane A (143)	In vitro	20 µM	[45]
	Flemichin D (**139**)	In vitro	20 µM	[45]
	6,8-Diprenyl-7,4′-dihydroxyflavanone (**119**)	In vitro	IC_50_ *=* 12.21 µM	[45]
	Water extract of roots	In vivo	100 mg/kg	[13]
	Non-alkaloid extracts of roots	In vivo	400 mg/kg	[13]
Reduce IL- 6	2′-Hydroxyglabrol (**131**)	In vitro	IC_50_ *=* 1.62 µM	[47]
	Glabrol (**121**)	In vitro	IC_50_ *=* 0.73 µM	[47]
	Maackiain (**168**)	In vitro	IC_50_ *=* 3.01 µM	[47]
	Bolusanthin IV (**261**)	In vitro	IC_50_ *=* 4.02 µM	[47]
	Ethanol extract of roots	In vivo	100 mg/kg	[7]
	(−)-Anagyrine (**61**)	In vitro	50 µM	[12]
	Sophocarpine (**34**)	In vitro	50 µM	[12]
	14*β-*Hydroxymatrine (**28**)	In vitro	50 µM	[12]
	7*β*-Sophoramine (**49**)	In vitro	50 µM	[12]
	Matrine (**1**)	In vitro	50 µM	[12]
	(+)-5*α*-Hydroxyoxymatrine (**3**)	In vivo	50 µM	[12]
	(+)-5*α*-Hydroxymatrine (**5**)	In vivo	50 µM	[12]
	12,13-Dehydrosophoridine (**16**)	In vitro	50 µM	[23]
	13*α*-Hydroxymatrine (**36**)	In vitro	50 µM	[23]
	13*β*-Hydroxymatrine (**37**)	In vitro	50 µM	[23]
	Isosophocarpine (**48**)	In vitro	50 µM	[23]
	Sophoridine (**13**)	In vitro	50 µM	[23]
	Water extract of roots	In vivo	0.3 g/kg	[75]
Reduce IL-5	50% (*v/v*) ethanol-water mixture	In vivo	100 mg/kg	[76]
Reduce IL-10	Ethanol extract of roots	In vivo	100 mg/kg	[7]
Reduce IL-1*β*	Water extract of roots	In vivo	0.3 g/kg	[75]
Reduced the hyperplasia of goblet cell	50% (*v/v*) ethanol-water mixture	In vivo	10 mg/kg	[76]
Inhibit xylene induced auricle swelling in mice	Oxymatrine (**4**)	In vivo	40 mg/kg	[78]
	(−)-Cytisine (**50**)	In vivo	40 mg/kg	[78]
	*S. tonkinensis* particles	In vivo	1.75 g/kg	[79]
Inhibit pain induced by acetic acid stimulation of the celiac mucosa	Matrine (**1**)	In vivo	40 mg/kg	[78]
	Sophoridine (**13**)	In vivo	30 mg/kg	[78]
	Sophocarpine (**34**)	In vivo	40 mg/kg	[78]
	*S. tonkinensis* particles	In vivo	3.5 g/kg	[79]
Inhibit croton oil induced ear swelling in mice	Water extract of roots	In vivo	0.35–1.12 g/kg	[80]
	Ethanol extract of roots	In vivo	0.35–1.12 g/kg	[80]
	Water extract of roots	In vivo	0.39 g/kg	[81]
**Anti-tumor activity**				
Inhibit A549	(−)-*N*-hexanoylcytisine (**55**)	In vitro	IC_50_ *=* 31.64 µM	[24]
	(−)-*N*-Formylcytisine (**52**)	In vitro	IC_50_ *=* 22.05 µM	[24]
	(6a*R*, 11a*R*)-Maackiain (**168**)	In vitro	IC_50_ *=* 24.58 µM	[24]
	Water extracts of roots	In vitro	6.5 µg/µL	[82]
	1-(6,7-Dihydro-5H-pyrrolo [1,2-*a*] imidazol-3-yl) ethenone (**76**)	In vitro	IC_50_ *=* 23.05 ± 0.46 µM	[27]
Inhibit HL-60	Tonkinensisol (**93**)	In vitro	IC_50_ *=* 36.48 μg/mL	[15]
	Sophoranol (**5**)	In vitro	10.00 µg/mL	[83]
	13,14-Dehydrosophoranol (**24**)	In vitro	1.00 µg/m L	[83]
Inhibit HepG2	Tonkinensine C (**73**)	In vitro	IC_50_ *=* 87.4 ± 7.1 µM	[1]
	Perlolyrine (**74**)	In vitro	IC_50_ *=* 91.8 ± 3.5 µM	[1]
	Harmine (**72**)	In vitro	IC_50_ *=* 48.9 ± 5.2 µM	[1]
	Alkaloids	In vitro	IC_50_ *=* 9.04 g/L	[84]
	Non-alkaloids extract of roots	In vitro	IC_50_ *=* 0.98 g/L	[84]
	Water extracts of roots	In vitro	6.5 µg/µL	[82]
Inhibit SH-SY5Y	Sophoranone (**120**)	In vitro	IC_50_ = 18.49 µM	[85]
	Matrine (**1**)	In vitro	IC_50_ = 60.81 µM	[85]
	Oxymatrine (**4**)	In vitro	IC_50_ = 42.56 µM	[85]
	(−)-Trifolirhizin (**190**)	In vitro	IC_50_ = 72.11 µM	[85]
	(−)-Maackiain (**168**)	In vitro	IC_50_ = 65.62 µM	[85]
Inhibit B16-BL6	Extract of roots	In vitro	400 µg/mL	[86]
Inhibit CNE-1, CNE-2	Chloroform extract of roots	In vitro	25 µg/mL	[87]
Inhibit U937	Sophoranone (**120**)	In vitro	IC_50_ = 3.8 ± 0.9 µM	[88]
Inhibit HeLa	Tonkinensine B (**59**)	In vitro	IC_50_ *=* 24.3± 0.3 µM	[25]
Inhibit MDA-MB-231	Tonkinensine B (**59**)	In vitro	IC_50_ *=* 48.9± 0.5 µM	[25]
	Water extract of roots	In vitro	6.5 µg/µL	[82]
Inhibit ESC solid tumor cell	Total alkaloids of roots	In vivo	100 mg/kg	[89]
Inhibit H_22_ ascites tumor cells	Total alkaloids of roots	In vivo	100 mg/kg	[89]
Inhibit S_180_ solid tumor cell	Total alkaloids of roots	In vivo	75 mg/kg	[89]
Inhibit BV2 glioma cell lines	Sophotokin (**174**)	In vitro	10 µM	[34]
	Maackiain (**168**)	In vitro	10 µM	[34]
	Medicarpin (**176**)	In vitro	10 µM	[34]
Inhibit Hep3B and KG-1 cells	Water extract of roots	In vitro	6.5 µg/µL	[82]
Decrease the number of cancer nodules in tumor tissue and reduce AFP in serum	Alkaloids extract of roots	In vivo	0.036 g/kg	[90]
**Effects on the liver**				
Protect HepG2 cell against acetaminophen (APAP)- induced damage	4-Methoxybenzamide (**252**)	In vitro	10 µmol/L	[64]
	7,3’-Dihydroxy-8,4’-dimethoxyisoflavone (**104**)	In vitro	10 µmol/L	[64]
	7,4’-Dihydroxy-3’-methoxyisoflavone (**102**)	In vitro	10 µmol/L	[64]
	(±)-3-(*p*-Methoxyphenyl)-1,2-propanediol (**255**)	In vitro	10 µmol/L	[64]
Enhance L-02 hepatocytes	Matrine (**1**)	In vivo and vitro	10 µM	[91]
	Oxymatrine (4)	In vivo and vitro	10 µM	[91]
Increase SOD and GSH	Non-alkaloids extract of roots	In vivo	400 mg/kg	[13]
	Water extract of roots	In vivo	400 mg/kg	[13]
Increase ALT and AST	Water extract of roots	In vivo	0.59 g/kg	[92]
Increase CPT 1A activity	Water extract of roots	In vivo	25 μg/mL	[91]
Reduce nonestesterified fatty acid Induce cellular lipids accumulation in hepatocytes	Matrine (**1**)	In vivo	10 µM	[91]
	Oxymatrine (**4**)	In vivo	10 µM	[91]
Reduce immune liver injury	Oxymatrine (**4**)	In vivo	60 mg/kg	[93]
	Sophocarpine (**34**)	In vivo	60 mg/kg	[93]
	Oxymatrine (**4**)	In vivo	120 mg/kg	[94]
Inhibite acetaminophen-induced hepatic oxidative damage in mice	STRP1 (Polysaccharide part)	In vivo	200 mg/kg	[95]
	STRP2 (Polysaccharide part)	In vivo	200 mg/kg	[95]
Alleviate non-alcoholic fatty liver disease of mice	Water extract of roots	In vivo	90 mg/kg	[91]
Inhibit the production of tyrosinase	Formononetin-7-O-*β*-_D_-glucoside(**116**)	In vitro	IC_50_ *=* (7.82 ± 0.28) × 10^−4^ mol/L	[43]
	Tectorigenin (**112**)	In vitro	IC_50_ *=* (3.73 ± 0.45) × 10^−4^ mol/L	[43]
	8-Prenylkeamferol (**91**)	In vitro	IC_50_ *=* (1.58 ± 0.31) × 10^−5^ mol/L	[43]
Reduce AST and ALT	Oxymatrine (**4**)	In vivo	120 mg/kg	[93]
	Sophocarpine (**34**)	In vivo	120 mg/kg	[93]
	Water extract of roots	In vivo	0.25 g/kg	[96]
Reduce AST	Non-alkaloid extract of roots	In vivo	100 mg/kg	[13]
	Water extract of roots	In vivo	200 mg/kg	[13]
Reduce ALT	Non-alkaloid extracts of roots	In vivo	400 mg/kg	[13]
	Water extract of roots	In vivo	200 mg/kg	[13]
**Anti-viral activity**				
Anti-Coxsackie virus B3	(−)-12*β*-Hydroxyoxysophocarpine (**18**)	In vitro	IC_50_ *=* 26.62 µM	[14]
	(−)-9*α*-Hydroxysophocarpine (**25**)	In vitro	IC_50_ *=* 197.22 µM	[14]
	(+)-Sophoranol (**5**)	In vitro	IC_50_ *=* 252.18 µM	[14]
	(−)-14*β*-Hydroxymatrine (**28**)	In vitro	IC_50_ *=* 184.14 µM	[14]
	3-(4-Hydroxyphenyl)- 4- (3- methoxy- 4-hydroxyphenyl)-3,4-dehydroquinolizidine (**75**)	In vitro	IC_50_ *=* 6.40 µM	[26]
	Cermizine C (**70**)	In vitro	IC_50_ *=* 3.25 µM	[26]
	Jussiaeiine A (**68**)	In vitro	IC_50_ *=* 4.66 µM	[26]
	Jussiaeiine B (**67**)	In vitro	IC_50_ *=* 3.21 µM	[26]
	(+)-5α-Hydroxyoxysophocarpine (**17**)	In vitro	IC_50_ *=* 0.12 µM	[26]
	(−)-12β-Hydroxyoxysophocarpine (**18**)	In vitro	IC_50_ *=* 0.23 µM	[26]
	(−)-Clathrotropine (**64**)	In vitro	IC_50_ *=* 1.60 µM	[26]
Anti-tobacco mosaic virus (TMV)	Sophtonseedlin B (**188**)	In vitro	100 µg/mL	[56]
	(−)-Trifolirhizin (**190**)	In vitro	100 µg/mL	[56]
	Sophtonseedline B (**21**)	In vitro	100 µg/mL	[19]
	Sophtonseedline D (**23**)	In vitro	100 µg/mL	[19]
	Sophtonseedline F (**8**)	In vitro	100 µg/mL	[19]
	(−)-*N*-Formylcytisine (**52**)	In vitro	100 µg/mL	[19]
	Alkaloid extracts of seeds	In vitro	0.5 mg/mL	[19]
	Methanol extracts of seeds	In vitro	0.5 mg/mL	[19]
Anti-hepatitis B virus (HBV)	(+)-Oxysophocarpine (**20**)	In vitro	0.4 µmol/mL	[20]
	(−)-Sophocarpine (**34**)	In vitro	0.4 µmol/mL	[20]
	(+)-Lehmannine (**14**)	In vitro	0.4 µmol/mL	[20]
	(−)-13,14-Dehydrosophoridine (**16**)	In vitro	1.6 µmol/mL	[20]
	(−) -14β-Hydroxyoxymatrine (**6**)	In vitro	0.4 µmol/mL	[18]
	(+)-Sophoranol (**5**)	In vitro	0.2 µmol/mL	[18]
	(−)-Cytisine (**50**)	In vitro	0.2 µmol/mL	[18]
Anti-mouse hepatitis virus	Methanol extracts of plant	In vitro	EC_50_ *=* 27.5 ± 1.1 µg/mL	[97]
Inhibited influenza virus A/Hanfang/359/95	(+)-12α-Hydroxysophocarpine (**15**)	In vitro	IC_50_ *=* 84.70 µM	[14]
	(−)-12β-Hydroxysophocarpine (**19**)	In vitro	IC_50_ *=* 242.46 µM	[14]
	(+)-Sophoramine (**49**)	In vitro	IC_50_ *=* 63.07 µM	[14]
**Anti-oxidant capacity**				
ABTS free radical scavenging ability	Chloroform extract of roots	In vitro	EC_50_ = 1.08 mg/mL	[98]
	Ethyl acetate extract of roots	In vitro	EC_50_ = 0.55 mg/mL	[98]
	*N*-butanol extract of roots	In vitro	EC_50_ = 1.27 mg/mL	[98]
	Ethanol extract of roots	In vitro	EC_50_ = 3.08 mg/mL	[98]
	Shandougenines A (**263**)	In vitro	IC_50_ = 0.532 ± 0.076 mM	[40]
	Shandougenines B (**264**)	In vitro	IC_50_ = 0.18 ± 0.032 mM	[40]
	Bolusanthin IV (**261**)	In vitro	IC_50_ = 0.3 ± 0.025 mM	[40]
	2-(2′,4′-Dihydroxyphenyl)-5,6-methylenedioxybenzofuran (**260**)	In vitro	IC_50_ = 0.726 ± 0.041 mM	[40]
	Shandougenine C (**127**)	In vitro	IC_50_ = 0.382 ± 0.055 mM	[40]
	Shandougenine D (**128**)	In vitro	IC_50_ = 0.341 ± 0.058 mM	[40]
	Demethylmedicarpin (**179**)	In vitro	IC_50_ = 0.503 ± 0.036 mM	[40]
Scavenging of DPPH radicals	Ethyl acetate extract of roots	In vitro	0.5 mg/mL	[98]
	Ethanol extract of roots	In vitro	0.5 mg/mL	[98]
	Chloroform extract of roots	In vitro	0.5 mg/mL	[98]
	*N*-butanol extract of roots	In vitro	0.5 mg/mL	[98]
	Water extract of aerial parts	In vitro	IC_50_ = 0.1434 g/L	[17]
	*N*-butyl alcohol extract of aerial parts	In vitro	IC_50_ = 0.0754 g/L	[17]
	Ethyl acetate extract of aerial parts	In vitro	IC_50_ = 0.0693 g/L	[17]
	Dichloromethane of aerial parts	In vitro	IC_50_ = 0.0494 g/L	[17]
	Petroleum ether extract of aerial parts	In vitro	IC_50_ = 0.1218 g/L	[17]
	STRP1 (Polysaccharide part)	In vitro	1.0 mg/mL	[95]
	STRP2 (Polysaccharide part)	In vitro	1.0 mg/mL	[95]
	Tonkinensisol (**93**)	In vitro	IC_50_ = 0.616 ± 0.021 mM	[40]
	Bolusanthin IV (**261**)	In vitro	IC_50_ = 0.502 ± 0.101 mM	[40]
	2-(2′,4′-Dihydroxyphenyl)-5,6-methylenedioxybenzofuran (**260**)	In vitro	IC_50_ = 0.527 ± 0.054 mM	[40]
	Shandougenines A (**263**)	In vitro	IC_50_ = 1.213 ± 0.101 mM	[40]
	Shandougenines B (**264**)	In vitro	IC_50_ = 0.327 ± 0.022 mM	[40]
	WRSP-A2b (Polysaccharide part)	In vitro	IC_50_ = 19.95 ± 0.25 mg/mL	[99]
	WRSP-A3a (Polysaccharide part)	In vitro	IC_50_ = 5.99 ± 0.20 mg/mL	[99]
Reducing power	Chloroform extract of roots	In vitro	EC_50_ = 0.60 mg/mL	[98]
	Ethyl acetate extract of roots	In vitro	EC_50_ = 0.64 mg/mL	[98]
	*N*-butanol extract of roots	In vitro	EC_50_ = 0.51 mg/mL	[98]
	Ethanol extract of roots	In vitro	EC_50_ = 0.84 mg/mL	[98]
Hydroxyl radical scavenging ability	Chloroform extract of roots	In vitro	EC_50_ = 1.33 mg/mL	[98]
	Ethyl acetate extract of roots	In vitro	EC_50_ = 2.80 mg/mL	[98]
	*N*-butanol extract of roots	In vitro	EC_50_ = 5.00 mg/mL	[98]
	WRSP-A2b (Polysaccharide part)	In vitro	IC_50_ = 19.78 ± 0.47 mg/mL	[99]
	WRSP-A3a (Polysaccharide part)	In vitro	IC_50_ = 8.38 ± 0.18 mg/mL	[99]
Superoxide anion radical scavenging ability	WRSP-A2b (Polysaccharide part)	In vitro	IC_50_ = 4.24 ± 0.11 mg/mL	[99]
	WRSP-A3a (Polysaccharide part)	In vitro	IC_50_ = 1.94 ± 0.05 mg/mL	[99]
**Toxicity**				
Respiratory depression, muscle fibrillation, convulsions, spasms, and death	Hydroalcoholic extract from the roots	Mice (i.g.)	LD_50_ = 9.802 ± 2.0067 g/kg	[100]
Convulsions, hair erection, rapid abdominal contraction and excitement, depression, abdominal breathing and eye closure, and death	(−)- Cytisine (**50**)	Mice (i.g.)	LD_50_ = 48.16 mg/kg	[101]
Irritability, hyperactivity, shortness of breath, and convulsions	Water extract of roots	Mice (i.g.)	LD_50_ = 17.469 g/kg	[102]
	90% Ethanol extract of roots	Mice (i.g.)	LD_50_ = 27.135 g/kg	[102]
	Alkaloids of roots	Mice (i.g.)	LD_50_ = 13.399 g/kg	[102]
	Water and 70% Ethanol extract mixture of roots	Mice (i.g.)	MTD = 36 g/kg	[103]
	All-component of of roots	Mice (i.g.)	MTD = 10.68 g/kg	[102]
Slow heartbeat, bent trunk of zebrafish, accelerated movement frequency, and abnormal movement track, Hepato renal, pericardial enlargement, death.	Sophoranone (**120**)	Zebrafish (p.o.)	LC_50_ = 22.45 µmol/L	[104]
To cause hepatomegaly	Sophoranone (**120**)	Zebrafish (p.o.)	3.86 µmol/L	[104]
The zebrafish liver lost transparency and became dark or brown, and liver blood flow was no longer observable	Dealkalized water extract of roots	Zebrafish (p.o.)	LC_10_ = 1009.1 µg/mL	[105]
	Ethanol sedimentation extract of roots	Zebrafish (p.o.)	LC_10_ = 4367.6 µg/mL	[105]
	*N*-Butyl ethanol extract of roots	Zebrafish (p.o.)	MNLC = 700.0 µg/mL	[105]
Slowed heart rate, reduced blood flow, and absence of circulation in the cardiotoxic phenotype, neurotoxic, and presents with behavioral abnormalities, bent trunk.	Sophoranone (**120**)	Zebrafish (p.o.)	11.59 µmol/L	[104]
Induced pericardial edema and slowed the blood circulation, heart rate lower	Diethyl ether extract of roots	Zebrafish (p.o.)	LC_10_ = 93.6 µg/mL	[105]
	*N*-Butyl ethanol extract of roots	Zebrafish (p.o.)	LC_10_ = 538.3 µg/mL	[105]
Pericardial edema, a misshaped atrium and ventricle as well as reduced number of endothelial cells and cardiomyocytes	Dichloromethane extract of roots	Zebrafish (p.o.)	MNLC = 450.0 µg/mL	[105]
Delayed yolk sac resorption in the hepatotoxic phenotype and Intestinal dysplasia	Sophoranone (**120**)	Zebrafish (p.o.)	1.29 µmol/L	[104]
To cause renal and pericardial edema	Sophoranone (**120**)	Zebrafish (p.o.)	15.57 µmol/L	[104]
**Other pharmacological activities**				
Inhibit *Pseudomonas aeruginosa*	2’,4’,7-Trihydroxy-6,8-bis(3-methyl-2-butenyl) flavanone (**259**)	In vitro	MIC *=* 125.0 µg/mL	[16]
	Genistin (**115**)	In vitro	MIC *=* 15.6 µg/mL	[16]
Inhibit *Bacillus megaterium*	2-Methoxy-6-methyl-1,4-benzoquinone (**277**)	In vitro	MIC *=* 3.125 µg/mL	[65]
	Xylariphilone (**282**)	In vitro	MIC *=* 12.5 µg/mL	[65]
	Xylarphthalide A (**283**)	In vitro	MIC *=* 25 µg/mL	[67]
	(−)-5-Carboxylmellein (**280**)	In vitro	MIC = 25 µg/mL	[67]
	(−)-5-Methylmellein (**281**)	In vitro	MIC = 25 µg/mL	[67]
Inhibit *Escherichia coli*	Lanatine A (**65**)	In vitro	MIC *=* 1.0 g/L	[26]
	Jussiaeiines A (**68**)	In vitro	MIC *=* 3.2 g/L	[26]
	Jussiaeiines B (**67**)	In vitro	MIC *=* 0.8 g/L	[26]
	(−)-5-Carboxylmellein (**280**)	In vitro	MIC = 25 µg/mL	[67]
	21-Acetoxycytochalasin J_3_ (**304**)	In vitro	MIC = 12.5 µg/mL	[71]
	2-(2’,4’-Dihydroxy)-5,6-dioxomethylbenzofuran (**260**)	In vitro	MIC *=* 31.3 µg/mL	[16]
	Xylarphthalide A (**283**)	In vitro	MIC *=* 25 µg/mL	[67]
	(−)-5-Methylmellein (**281**)	In vitro	MIC = 25 µg/mL	[67]
	6-Heptanoyl-4-methoxy-2H-pyran-2-one (**286**)	In vitro	MIC *=* 50 µg/mL	[106]
Inhibit *Staphylococcus aureus*	3-(4-Hydroxyphenyl)-4-(3-methoxy-4-hydroxyphenyl) -3,4-dehydroquinolizidine (**75**)	In vitro	MIC *=* 8.0 g/L	[26]
	Cermizines C (**70**)	In vitro	MIC *=* 3.5 g/L	[26]
	Jussiaeiines B (**67**)	In vitro	MIC *=* 6.0 g/L	[26]
	Cytochalasin K (**311**)	In vitro	MIC *=* 12.5 µg/mL	[65]
	6-Heptanoyl-4-methoxy-2H-pyran-2-one (**286**)	In vitro	MIC *=* 50 µg/mL	[106]
	(−) -*N*-methylcytisine (**54**)	In vitro	MIC *=* 12.0 g/L	[26]
	Xylarphthalide A (**283**)	In vitro	MIC *=* 25 µg/mL	[67]
	(−)-5-Carboxylmellein (**280**)	In vitro	MIC = 25 µg/mL	[67]
	(−)-5-Methylmellein (**281**)	In vitro	MIC = 12.5 µg/mL	[67]
	Cytochalasin K (**311**)	In vitro	MIC *=* 12.5 µg/mL	[65]
	2’,4’,7-Trihydroxy-6,8-bis(3-methyl-2-butenyl) flavanone (**259**)	In vitro	MIC *=* 62.5 µg/mL	[16]
	Ethyl acetate extract of roots	In vitro	MIC = 0.313 mg/mL	[98]
Inhibit *Shigella dysenteriae*	Xylarphthalide A (**283**)	In vitro	MIC *=* 25 µg/mL	[67]
	(−)-5-Methylmellein (**281**)	In vitro	MIC = 25 µg/mL	[67]
	(−)-3-Carboxypropyl-7-hydroxyphthalide (**293**)	In vitro	MIC = 12.5 µg/mL	[69]
Inhibit *Proteus vulgaris*	Xylareremophil (**287**)	In vitro	MIC *=* 25 µg/mL	[68]
	Mairetolide G (**291**)	In vitro	MIC = 25 µg/mL	[68]
Inhibit *Micrococcus luteus*	Mairetolide G (**291**)	In vitro	MIC = 50 µg/mL	[68]
	Mairetolide B (**290**)	In vitro	MIC = 50 µg/mL	[68]
	Xylareremophil (**287**)	In vitro	MIC *=* 25 µg/mL	[68]
Inhibit *Micrococcus lysodeikticus*	Mairetolide B (**290**)	In vitro	MIC = 100 µg/ml	[68]
	Mairetolide G (**291**)	In vitro	MIC = 100 µg/mL	[68]
	Xylareremophil (**287**)	In vitro	MIC *=* 100 µg/mL	[68]
Inhibit *Bacillus subtilis*	(−)-5-Carboxylmellein (**280**)	In vitro	MIC = 12.5 µg/mL	[67]
	Mairetolide B (**290**)	In vitro	MIC = 100 µg/mL	[68]
	Mairetolide G (**291**)	In vitro	MIC = 100 µg/mL	[68]
	Xylarphthalide A (**283**)	In vitro	MIC *=* 25 µg/mL	[67]
	(−)-5-Methylmellein (**281**)	In vitro	MIC = 12.5 µg/mL	[67]
	Xylapeptide A (**301**)	In vitro	MIC = 12.5 µg/mL	[70]
	(−)-3-Carboxypropyl-7-hydroxyphthalide (**293**)	In vitro	MIC = 25 µg/mL	[69]
	Xylareremophil (**287**)	In vitro	MIC *=* 100 µg/mL	[68]
Inhibit *Bacillus anthracis*	(−)-5-Carboxylmellein (**280**)	In vitro	MIC = 25 µg/mL	[67]
	21-Acetoxycytochalasin J_3_ (**304**)	In vitro	MIC = 12.5 µg/mL	[71]
Inhibit *Alternaria oleracea*	Cytochalasin E (**310**)	In vitro	MIC = 3.125 µg/mL	[71]
	Cytochalasin H (**306**)	In vitro	MIC = 6.25 µg/mL	[71]
Inhibit *Colletotrichum capsici*	Cytochalasin E (**310**)	In vitro	MIC = 1.56 µg/mL	[71]
	Cytochalasin H (**306**)	In vitro	MIC = 6.25 µg/mL	[71]
Inhibit *Pestalotiopsis theae*	Cytochalasin E (**310**)	In vitro	MIC = 1.56 µg/mL	[71]
	Cytochalasin H (**306**)	In vitro	MIC = 12.5 µg/mL	[71]
Inhibit *Enterobacter areogenes*	(−)-3-Carboxypropyl-7-hydroxyphthalide methyl ester (**294**)	In vitro	MIC = 12.5 µg/mL	[69]
	(−)-3-Carboxypropyl-7-hydroxyphthalide (**293**)	In vitro	MIC = 12.5 µg/mL	[69]
Inhibit *Colletotriehum gloeosporioides*	Methanol extract of roots	In vitro	EC_50_ = 1.214 mg/mLMIC = 2.5 mg/mL	[107]
Inhibit *Fusarium solani*	Methanol extract of roots	In vitro	EC_50_ = 1.169 mg/mLMIC = 2.5 mg/mL	[107]
Inhibit *Ceratocystis paradoxa*	Cytochalasin H (**306**)	In vitro	MIC = 25 µg/mL	[71]
Inhibit *Bacillus cereus*	Xylapeptide A (**301**)	In vitro	MIC = 12.5 µg/mL	[70]
Moderate activities against *Aphis fabae*	Sophtonseedline G (**9**)	In vivo	LC_50_ *=* 38.29 mg/L	[19]
	Matrine (**1**)	In vivo	LC_50_ *=* 18.63 mg/L	[19]
	(−)-*N*-Formylcytisine (**52**)	In vivo	LC_50_ *=* 23.74 mg/L	[19]
Decreased fasting blood glucose levels	Matrine (**1**)	In vivo	2.5 mg/kg	[108]
	Ethyl acetate extract of roots	In vivo	60 mg/kg	[33]
alleviate insulin resistance	Ethyl acetate extract of roots	In vivo	60 mg/kg	[33]
	Matrine (**1**)	In vivo	10 mg/kg	[108]
Inhibit 5-lipoxygenase	50 % (*v/v*) Ethanol–water mixture	In vitro	IC_50_ = 1.61 µg/mL	[76]
	Maackiain (**168**)	In vitro	IC_50_ = 7.9 µM	[76]
	Sophoranone (**120**)	In vitro	IC_50_ = 1.6 µM	[76]
Inhibit thromboxane synthase	50 % (*v/v*) Ethanol–water mixture	In vitro	IC_50_ = 5.56 µg/mL	[76]
Inhibit butyrylcholinesterase	Ethanol extract of roots	In vitro	IC_50_ = 15. 169 µg/mL	[109]

## Data Availability

Not applicable.

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
