# Peer review of "Biological Activities and Secondary Metabolites from Sophora tonkinensis and Its Endophytic Fungi"

_molecules, 2022, doi:10.3390/molecules27175562_

Round 1

Reviewer 1 Report

Accept in the current form.

Author Response

Thanks for your recommendation.

Reviewer 2 Report

I would like to ask you to change the topic Phytochemistry into phytochemistry and mycochemistry as it included chemistry of endophytic fungi and fungi are not plants any more

Author Response

Thanks very much for your carefully review. If possible, we will modify the topic as required.

Reviewer 3 Report

In this manuscript of "Biological Activities and Secondary Metabolites from Sophora tonkinensis and its Endophytic Fungi", the authors presented 316 compounds, which were isolated from S. tonkinensis and its endophytic fungi. Some of the identified molecules exhibited good biological activities, which supported the usage of S. tonkinensis in the folk treatment of upper respiratory tract infection diseases.

However, the following concerns should be addressed for further publishing consideration.

1. I hope the author could split the Table 1 and Table 2 into several pieces, based on the different structures and pharmacological activities. The compound structures also are illustrated with the tables for easy understanding.

2. All the structure-activity relationship should be discussed one by one for details

3. The translation for the clinical and marketing Chinese patent medicines should be an issue. The "Kai Hou Jian Pen Wu Ji", "Gui Lin Xi Gua Shuang" should be "Jin Hou Jian throat spray", and "Watermelon Frost Spray". Please double check and correct.

4. All the S. tonkinensis should be italicized.

5. The numbers in R1 and R2 should be supscripted in Figure 1.

Author Response

1. Comment: I hope the author could split the Table 1 and Table 2 into several pieces, based on the different structures and pharmacological activities. The compound structures also are illustrated with the tables for easy understanding.

Authors′ response: Thanks a lot for your kind reminding and suggestion. For the continuity of the data and information transmited to readers, we did not split the Table 1 and Table 2 into several pieces.

2. Comment: All the structure-activity relationship should be discussed one by one for details.

Authors′ response: Thank you very much for your careful review, we totally agree with your opinion. However, the existing data is difficult to support the unquestionable structure-activity relationship discussion. Thus we did not add the structure-activity relationship discussion with lacking evidence.

3. Comment: The translation for the clinical and marketing Chinese patent medicines should be an issue. The "Kai Hou Jian Pen Wu Ji", "Gui Lin Xi Gua Shuang" should be "Jin Hou Jian throat spray", and "Watermelon Frost Spray". Please double check and correct.

Authors′ response: Thank you very much for your careful review. As suggested, we have corrected the name of the clinical and marketing Chinese patent medicines, which was highlighted in red in the manuscript-R1.

4. Comment: All the S. tonkinensis should be italicized.

Authors′ response: Many thanks for your careful review, in this paper, we checked the Latin names of the S. tonkinensis, and all changed in italic forms.

5. Comment: The numbers in R1 and R2 should be supscripted in Figure 1.

Authors′ response: Many thanks for your suggestions. We have checked the structures of Figure 1, and all the numbers in R of structures have been supscripted.

Round 2

Reviewer 3 Report

Now the manuscript could be accepted in its current version.